# FiLo: Zero-Shot Anomaly Detection by Fine-Grained Description and High-Quality Localization

Zhaopeng Gu[*]
Institute of Automation, Chinese Academy of Sciences
Beijing, China
School of Artificial Intelligence, University of Chinese Academy of Sciences
Beijing, China
guzhaopeng2023@ia.ac.cn

Bingke Zhu[*]
Institute of Automation, Chinese Academy of Sciences
Beijing, China
Objecteye Inc.
Beijing, China
bingke.zhu@nlpr.ia.ac.cn

Guibo Zhu[†]
Institute of Automation, Chinese Academy of Sciences
Beijing, China
University of Chinese Academy of Sciences
Beijing, China
gbzhu@nlpr.ia.ac.cn

Yingying Chen[†]
Institute of Automation, Chinese Academy of Sciences
Beijing, China
Objecteye Inc.
Beijing, China
yingying.chen@nlpr.ia.ac.cn

Hao Li[‡]
Central South University
Hunan, China
8209210109@csu.edu.cn

Ming Tang
Institute of Automation, Chinese Academy of Sciences
Beijing, China
University of Chinese Academy of Sciences
Beijing, China
tangm@nlpr.ia.ac.cn

Jinqiao Wang
Institute of Automation, Chinese Academy of Sciences
Beijing, China
University of Chinese Academy of Sciences
Beijing, China
Objecteye Inc.
Beijing, China
jqwang@nlpr.ia.ac.cn

## Abstract

Zero-shot anomaly detection (ZSAD) methods detect anomalies without prior access to known normal or abnormal samples within target categories. Existing methods typically rely on pretrained multimodal models, computing similarities between manually crafted textual features representing "normal" or "abnormal" semantics and image patch features to detect anomalies. However, the generic descriptions of "abnormal" often fail to precisely match diverse types of anomalies across different object categories. Additionally, computing feature similarities for single patches struggles to pinpoint specific locations of anomalies with various sizes and scales. To address these issues, we propose a novel ZSAD method called FiLo, comprising two components: adaptively learned **Fi**ne-**G**rained **Des**cription (FG-Des) and position-enhanced **H**igh-**Q**uality **Loc**alization (HQ-Loc). FG-Des introduces fine-grained anomaly descriptions for each category using Large Language Models (LLMs)

and employs adaptively learned textual templates to enhance the accuracy and interpretability of anomaly detection. HQ-Loc, utilizing Grounding DINO for preliminary localization, position-enhanced text prompts, and Multi-scale Multi-shape Cross-modal Interaction (MMCI) module, facilitates more accurate localization of anomalies of different sizes and shapes. Experimental results on datasets like MVTec and VisA demonstrate that FiLo significantly improves the performance of ZSAD in both detection and localization, achieving state-of-the-art performance with an image-level AUC of 83.9% and a pixel-level AUC of 95.9% on the VisA dataset. Code is available at https://github.com/CASIA-IVA-Lab/FiLo.

## CCS Concepts

• **Computing methodologies** → **Visual inspection**; *Natural language processing*; • **Information systems** → **Multimedia and multimodal retrieval**.

## Keywords

Vision-Language Model, Anomaly Detection, Zero-shot Learning

**ACM Reference Format:**
Zhaopeng Gu, Bingke Zhu, Guibo Zhu, Yingying Chen, Hao Li, Ming Tang, and Jinqiao Wang. 2024. FiLo: Zero-Shot Anomaly Detection by Fine-Grained Description and High-Quality Localization. In *Proceedings of the 32nd ACM International Conference on Multimedia (MM '24), October 28-November 1, 2024, Melbourne, VIC, Australia.* ACM, New York, NY, USA, 9 pages. https://doi.org/10.1145/3664647.3680685

[*]Both authors contributed equally to this research.
[†]Corresponding author.
[‡]Work done as intern in Institute of Automation, Chinese Academy of Sciences.

*MM '24, October 28-November 1, 2024, Melbourne, VIC, Australia*
© 2024 Copyright held by the owner/author(s).
ACM ISBN 979-8-4007-0686-8/24/10
https://doi.org/10.1145/3664647.3680685

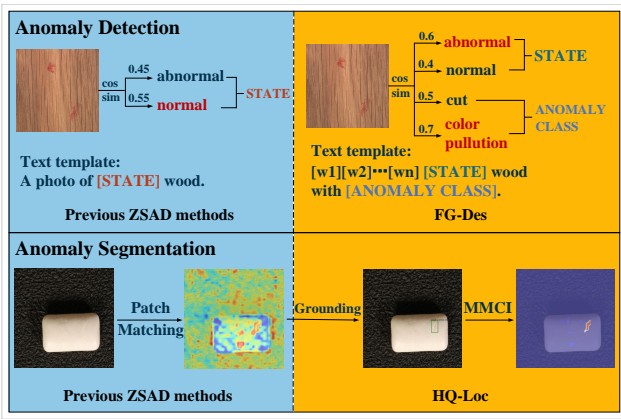

**Figure 1: Comparison of anomaly detection and localization between FiLo and previous ZSAD methods. Previous ZSAD methods utilize fixed templates and generic anomaly descriptions, potentially resulting in errors. Our FG-Des enhances detection accuracy with adaptively learned text templates and fine-grained anomaly descriptions. For localization, ZSAD methods often produce false positives in background areas by directly comparing image patches with text features. Our HQ-Loc approach, using Grounding DINO, location enhancement, and MMCI, effectively removes background regions and improves localization accuracy.**

## 1 Introduction

The anomaly detection task aims to identify whether industrial products contain abnormalities or defects and locate the abnormal regions within the samples, which plays a crucial role in product quality control and safety monitoring. Traditional anomaly detection methods [6, 7, 31, 35] typically require a large number of normal samples for model training. While performing well in some scenarios, they often fail in situations requiring protection of user data privacy or when applied to new production lines. Zero-Shot Anomaly Detection (ZSAD) has emerged as a research direction tailored to such scenarios, aiming to perform anomaly detection tasks effectively without prior data on the target item categories, demanding high generalization ability from the model.

Multimodal pre-trained models [18, 19, 29] have recently demonstrated strong zero-shot recognition capabilities in various visual tasks. Many works have sought to leverage the vision-language comprehension ability of multimodal pre-trained models for ZSAD tasks, such as WinCLIP [17], APRIL-GAN [5], and AnomalyGPT [16]. These methods assess whether an image contains anomalies by computing the similarity between image features and manually crafted textual features representing "normal" and "abnormal" semantics. They also localize abnormal regions by calculating the similarity between the image patch features and the textual features. While these approaches partly address the challenges of ZSAD, they encounter issues in both anomaly detection and localization. The generic "abnormal" descriptions fail to precisely match the diverse types of anomalies across different object categories. Moreover, computing feature similarity for individual patches struggles to precisely locate abnormal regions of varying sizes and shapes. To

tackle these issues, we propose FiLo (**Fi**ne-Grained Description and High-Quality **Lo**calization), which addresses the shortcomings of existing ZSAD methods through adaptively learned Fine-Grained Description (FG-Des) and High-Quality Localization (HQ-Loc), as depicted in Figure 1.

Concerning anomaly detection, manually crafted abnormal descriptions typically employ generic terms such as "damaged" or "defect" [5, 16, 17], which do not adequately capture the specific types of anomalies present across different object categories. Furthermore, existing methods' text prompt templates like *A xxx photo of xxx.* are primarily designed for foreground object classification tasks and may not be suitable for identifying normal and abnormal parts within objects. In FG-Des, we first leverage the capabilities of Large Language Models (LLMs) to generate fine-grained anomaly types for each object category, replacing generic abnormal descriptions with specific anomaly content that matches the anomaly samples better. Next, we utilize learnable text vectors instead of manually crafted sentence templates and embed the detailed anomaly content generated in the previous step into the adaptively learned text templates to improve the match between the text and the abnormal images, enhancing the textual features for anomaly detection. Our FG-Des not only improves the accuracy of anomaly detection but also enables the identification of the specific anomaly categories present in the samples, thus enhancing the interpretability.

Regarding anomaly localization, existing methods [5, 8, 16] localize anomalies by computing the similarity between the features of each image patch and the textual features. However, anomalies often span multiple patches with different shapes and sizes, sometimes requiring comparison with surrounding normal regions to determine their abnormality. While WinCLIP [17] addresses this issue by employing windows of different sizes, it incurs significant time and space costs by inputting a large number of images corresponding to each window into CLIP's image encoder during inference. To tackle this problem, we design HQ-Loc, which consists of three main components: first, preliminary anomaly localization based on Grounding DINO [22]. Considering that even in abnormal samples, most regions are normal, and anomalies only exist in small local areas, we utilize the detailed anomaly descriptions generated in the previous step and employ Grounding DINO [22] for preliminary anomaly localization. Although directly using Grounding DINO for zero-shot anomaly localization yields low accuracy, the localized regions are always in the foreground, effectively avoiding false positives in background regions. Second, position enhancement involves adding the position detected by Grounding DINO to the text prompt, resulting in a more accurate description of the anomaly position. Third, the Multi-scale Multi-shape Cross-modal Interaction (MMCI) module aggregates patch features extracted by the Image Encoder using convolutional kernels of different sizes and shapes to enhance the method's ability to localize anomalies of different sizes and shapes.

Extensive experiments are conducted on multiple datasets like MVTec [2] and VisA [39]. Our FiLo improves the accuracy of anomaly detection and localization, achieving new state-of-the-art zero-shot performance. Trained on the MVTec dataset and tested on the VisA dataset, FiLo achieves an image-level AUC of 83.9% and a pixel-level AUC of 95.9%, outperforming other ZSAD methods.

Our contributions can be summarized as follows:

- We propose an adaptively learned Fine-Grained Description (FG-Des) that leverages domain-specific knowledge to introduce detailed anomaly descriptions, replacing generic "normal" and "abnormal" descriptions. Also, we use learnable vectors instead of manually crafted text templates to learn textual content which is more suitable for anomaly detection, improving both the accuracy and interpretability.
- Additionally, we design a High-Quality Localization method (HQ-Loc) that employs Grounding DINO [22] for preliminary anomaly localization, enhances text prompts with descriptions of anomaly positions, and utilizes an MMCI module to localize anomalies of different sizes and shapes more accurately, improving anomaly localization accuracy.
- Extensive experiments on multiple datasets demonstrate significant performance improvements in anomaly detection and localization compared to baseline methods. FiLo has been proved to be effective for zero-shot anomaly detection and localization, achieving state-of-the-art performance.

## 2  Related work

### 2.1  Vision-Language Models

Recently, multimodal models integrating visual and textual content have achieved significant success in various visual tasks [3, 19, 22, 29]. Among these, CLIP [29], pre-trained on a massive scale internet dataset, emerges as one of the most prominent methods. CLIP employs two structurally similar Transformer [34] encoders to extract features from images and text, aligning features with the same semantics through contrastive learning methods. With appropriate prompts, CLIP demonstrates remarkable zero-shot generalization capabilities across multiple datasets for downstream image classification tasks. However, the quality of prompts significantly affects the performance of downstream tasks. Traditional approaches [4, 17] require experts to manually craft suitable text prompts for each task, demanding domain-specific knowledge and being time-consuming. Recent methods like coop [37] and cocoop [36] propose using learnable vectors instead of manually crafted prompts, requiring minimal training cost while achieving superior performance across multiple datasets.

While the original CLIP was designed for image classification tasks, researchers have extended their efforts to explore vision-language models for object detection and semantic segmentation tasks. Grounding DINO [22] is a notable example, combining the Transformer-based object detector DINO with Grounded pretraining, achieving excellent performance as an open-set object detector.

Our FG-Des method, incorporating adaptive learned fine-grained anomaly descriptions, is built upon CLIP [29] and cocoop [36]. However, straightforward utilization of cocoop-enhanced CLIP does not excel in anomaly detection tasks. Detailed anomaly descriptions for each item category are crucial for achieving outstanding performance. Grounding DINO [22] serves as a vital component of HQ-Loc. Yet, employing Grounding DINO [22] directly for zero-shot anomaly localization yields low accuracy. We utilize Grounding DINO solely for preliminary anomaly localization, capturing the approximate location of anomalies and avoiding false positives in background regions.

### 2.2  Zero-shot Anomaly Detection

Most zero-shot anomaly detection methods leverage the transferability of pre-trained vision-language models. Early methods like ZoC [13] and CLIP-AD [23], simply apply CLIP to anomaly detection data, resulting in low accuracy and inability to localize abnormal regions. WinCLIP [17] first achieves anomaly localization by cropping windows of different sizes in images and significantly enhances anomaly detection by employing carefully crafted text prompts. APRIL-GAN [5] aligns patch-level image features with textual features using a learnable linear projection layer to accomplish anomaly localization, overcoming the inefficiency caused by WinCLIP's input of numerous windows and further enhancing performance. AnoVL [8] resolves the mismatch between patch-level image features and textual features by introducing V-V attention [20], enabling direct application of CLIP to anomaly detection tasks without any additional training. However, all the above methods require carefully designed and manually crafted text templates. AnomalyCLIP [38], an emerging approach, substitutes object-agnostic learnable text vectors for manually crafted text templates. Nevertheless, AnomalyCLIP describes anomalies uniformly using the word "damaged", which is evidently insufficient to cover all types of anomalies.

SAA [4] is a zero-shot anomaly localization method based on the Grounded-SAM [30] approach. SAA utilizes Grounding DINO to generate anomaly bounding boxes, which are then used as prompts input into the Segment Anything Model [18] to obtain anomaly localization results. However, SAA [4] requires expertly crafted text inputs for Grounding DINO, and its results heavily rely on the detection outcomes of Grounding DINO, which may lead to low precision when directly applied to ZSAD. In our method, Grounding DINO serves solely as a preliminary anomaly localization module, aiming to prevent false positives in background regions of images. The primary dependency of our approach lies in the MMCI module for anomaly localization.

Moreover, none of the above methods incorporate location information of anomalies in the text prompt. Compared to existing methods, our approach enhances anomaly detection performance and interpretability by adaptive learned Fine-Grained anomaly Descriptions. We also improve the localization capability for anomalies of different sizes and shapes through our position-enhanced High-Quality localization method HQ-Loc.

### 2.3  Visual Description Enhancement

As mentioned earlier, numerous prior studies [36, 37] have extensively demonstrated that the quality of the text prompt significantly impacts the performance of downstream tasks for pretrained Vision-Language models like CLIP [29]. In contrast to text content meticulously crafted by experts, recent works [14, 25, 26] have delegated the task of generating high-quality text prompts to large language models (LLMs), which are called visual description enhancement. LLMs such as GPT-3.5 [28] and GPT-4 [1] encapsulate extensive knowledge across various domains, showcasing impressive performance across a spectrum of tasks. Our FiLo method harnesses the profound domain knowledge embedded within LLMs to generate potential anomaly types for each item category, thereby deriving

fine-grained anomaly descriptions. We are the first to apply visual description enhancement techniques to anomaly detection tasks.

## 2.4 Multi-Scale Convolution

In recent years, multi-scale convolution has been a research hotspot to detect objects of different sizes appearing in images [10, 11, 32, 33]. Multi-scale convolution methods aggregate features of regions with different sizes by using convolutional kernels of various sizes, achieving significant performance improvements in image classification, semantic segmentation, and object detection. InceptionNet [32] is a typical representative, simultaneously employing convolutional kernels of $1 \times 1$, $3 \times 3$, $5 \times 5$, etc. within the same layer to address the uncertainty of the optimal kernel size across different samples. MixConv [33] groups input channels and applies convolutional kernels of different sizes to each channel group. RepVGG [11] decomposes all sizes of convolutional kernels into a series of composite operations of $3 \times 3$ convolutions. ACNet [10] changes the order of convolution and summation, first summing convolutional kernels of different sizes and then performing a single convolution operation, thereby reducing computational overhead. Most existing multi-scale methods focus on square convolutional kernels of different sizes. ACNet [10] employs multi-shape convolutional kernels, but its emphasis is on computational efficiency, neglecting multi-scale aspects. Since anomalies in images may exhibit various shapes and sizes, our MMCI module introduces convolutional kernels of different sizes and shapes to fully localize anomalies.

## 3 FiLo

In this paper, we propose FiLo to enhance the capability of zero-shot anomaly detection and localization. Regarding anomaly detection, we devise the adaptively learned Fine-Grained Description method (FG-Des, Sec 3.2), which leverages fine-grained anomaly descriptions generated by LLMs and adaptable text vectors to identify the most precise textual representation for each anomaly sample. FG-Des facilitates more accurate judgments regarding the presence of anomalies in images and determines detailed anomaly types, thereby enhancing the interpretability of the method. For anomaly localization, we introduce the position-enhanced High-Quality Localization method (HQ-Loc, Sec 3.3), which employs preliminary localization via Grounding DINO, position-enhanced text prompts, and a Multi-scale, Multi-shape Cross-modal Interaction module to more accurately pinpoint anomalies of various sizes and shapes.

## 3.1 Overall Architecture

The overall architecture of the model is illustrated in Figure 2. For an input image $I \in \mathbb{R}^{H \times W \times 3}$, we first utilize information from the dataset or LLM to generate a list of fine-grained anomaly types that may exist for this item category. Subsequently, the anomaly text is inputted into Grounding DINO to obtain preliminary bounding boxes for anomaly localization. Simultaneously, the combination of fine-grained anomaly type and previously learned text vector templates yields text descriptions for both normal and abnormal cases. These descriptions are then fed into the CLIP Text Encoder for feature extraction, resulting in representations of normal and abnormal text features. Next, the image is passed through the CLIP Image Encoder to extract intermediate patch features $P_i \in \mathbb{R}^{H_i \times W_i \times C_i}$

from M stages, where $i$ indicates the $i$-th stage. These intermediate patch features are subjected to the MMCI module together with text features to generate anomaly map for each layer $M_i \in \mathbb{R}^{H \times W}$. Subsequently, after filtering with bounding boxes, the score maps for each layer are summed and normalized to obtain the final anomaly map $M \in \mathbb{R}^{H \times W}$. The global features of the image are compared with text features after adaptation, and the maximum value of the final anomaly map $M$ is added to derive the global anomaly score for the image.

## 3.2 FG-Des

Numerous existing methods [5, 8, 17] have demonstrated that the quality of text prompts significantly affects the effectiveness of anomaly detection when performing zero-shot inference on new categories. Therefore, we first focus on prompt engineering to generate more accurate and efficient text prompts for enhancing anomaly detection in ZSAD. In FG-Des, we achieve this goal through adaptively learned text templates and fine-grained anomaly descriptions generated by LLMs.

*3.2.1 Adaptively Learned Text Templates.* Following the success of methods like WinCLIP [17], subsequent methods such as APRIL-GAN [5] and AnomalyGPT [16] directly adopt the text templates used in WinCLIP to construct text prompts. However, the text template in WinCLIP, *A xxx photo of [state] [class]*, is primarily derived from the text template used by CLIP for image classification tasks on the ImageNet [9] dataset, which mainly indicates the category of foreground objects in the image rather than whether the object contains anomalies internally. To address this issue, we employ adaptive text templates learned based on anomaly detection-related data. During the learning process, these templates can combine the normal and abnormal content in the image to generate text prompts that better distinguish between normal and abnormal cases, while avoiding the need for extensive manual template engineering. Our adaptive normal and abnormal text templates are defined as follows:

$$T_n = [V_1][V_2]...[V_n][STATE][CLASS].$$
$$T_a = [W_1][W_2]...[W_n][STATE][CLASS]$$
$$with \ [ANOMALY \ CLASS] \ at \ [POS].$$

$[V_i]$ and $[W_i]$ are learnable text vectors, $[STATE]$ represents the general "normal" or "abnormal" state, $[CLASS]$ denotes the item category, $[ANOMALY \ CLASS]$ specifies the detailed anomaly content, and $[POS]$ indicates the location of the anomaly region, which can be one of nine possible scenarios, e.g., "top left" or "bottom".

Based on this template, we only need to replace the $[CLASS]$, $[ANOMALY \ CLASS]$, and $[POS]$ parts for different objects to generate different text prompt content.

*3.2.2 Fine-Grained Anomaly Descriptions.* As mentioned earlier, the generic "anomaly" texts in existing methods are insufficient to accurately describe the diverse types of anomalies that may appear on different object categories. Therefore, there is an urgent need for more personalized, informative text prompts to accurately characterize each image. LLMs such as GPT-4 [1] possess rich expert knowledge across various domains. We harness the power of LLMs to generate specific lists of potential anomaly types for each item category, replacing the vague and general "anomaly" or "damaged"

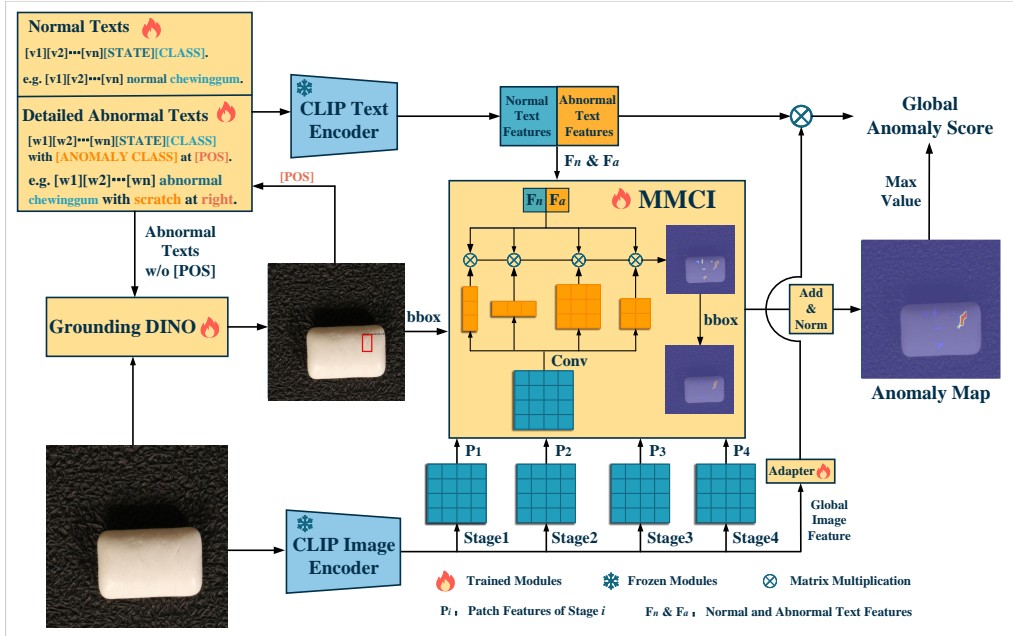

**Figure 2: Overall architecture of FiLo. Given an input image, fine-grained anomaly types are generated by LLM. Then normal and detailed abnormal texts are input into Grounding DINO to obtain bounding boxes and are fed into CLIP Text Encoder to get $F_n$ and $F_a$. Intermediate patch features of input image are subjected to MMCI together with text features to compute anomaly map, and the global image features are compared with text features after adaptation to obtain global anomaly score.**

descriptions used in previous methods. Such detailed textual features, when combined with features extracted by CLIP from images, lead to better anomaly detection results.

By incorporating fine-grained anomaly descriptions generated by large language models (LLMs) into the adaptive text templates' [ANOMALY CLASS] section, we obtain complete text prompts. These prompts are then inputted into the CLIP Text Encoder, and after group averaging, we obtain text features representing normal and abnormal cases, denoted as $F = [F_n, F_a] \in \mathbb{R}^{2 \times C}$. For the global features $G$ extracted from the image via the CLIP Image Encoder, we first pass them through a linear adapter layer to obtain adapted image features $A \in \mathbb{R}^C$ that better match the textual content. Next, we calculate the global anomaly score by Eq (1):

$$S_{global} = softmax(A \cdot F_a^T) + max(M). \tag{1}$$

$M$ represents the anomaly map calculated in Sec 3.3 and $max(\cdot)$ denotes the maximum operation.

Fine-grained anomaly descriptions not only improve the accuracy of anomaly detection but also enhance the interpretability of the detection results. Specifically, we can calculate the similarity between image features and each precise anomaly description. By examining the textual descriptions with high similarity, we can determine which category the anomaly in the image belongs to, thus gaining deeper insight into the model's decision-making process.

## 3.3 HQ-Loc

Existing Zero-Shot Anomaly Detection (ZSAD) methods often locate anomaly positions by computing the similarity between the

features of each image patch and textual features. However, an anomaly region often spans multiple patches, exhibiting various positions, shapes, and sizes. Sometimes, it requires comparison with surrounding normal regions to determine if it's an anomaly. To address this, we propose this position-enhanced High-Quality Localization method HQ-Loc, which enhances anomaly localization from coarse to fine. This is achieved through three key components: Grounding DINO preliminary localization, position-enhanced textual prompts, and Multi-Scale Multi-Shape Cross-modal Interaction Module (MMCI). Below, we provide detailed explanations for each component.

*3.3.1 Grounding DINO Preliminary Localization.* Existing ZSAD methods typically lack discrimination between patches at different positions in the image, often resulting in the misidentification of background perturbations as anomalies. To mitigate this, we utilize detailed anomaly descriptions generated in the previous step to perform preliminary anomaly localization using Grounding DINO. While direct application of Grounding DINO may not precisely determine the exact location of anomalies, the localization boxes obtained generally reside in the foreground of objects, often near the anomaly area. Therefore, using the localization results from Grounding DINO to restrict anomaly regions effectively avoids false positives in the background, thus enhancing the accuracy of anomaly localization. Additionally, since Grounding DINO localization is not entirely accurate and may have missed detections, we adopt a strategy of suppressing anomaly scores outside all localization boxes by multiplying them with a parameter $\lambda$.

*3.3.2 Position-Enhanced Textual Prompt.* After obtaining the preliminary anomaly localization results from Grounding DINO, we incorporate the position information from the localization boxes into textual prompts to enhance position descriptions. Textual prompts with detailed anomaly descriptions and position enhancements are more aligned with the content in the image being examined. This alignment assists the model in concentrating on specific areas of the image during anomaly localization in the subsequent step, thereby improving localization accuracy.

*3.3.3 MMCI Module.* To comprehensively locate anomalies of different shapes and sizes, our approach does not directly compute the similarity between each image patch feature and textual features. Instead, we design a Multi-Scale Multi-Shape Cross-Modal Interaction Module (MMCI). MMCI is inspired by WinCLIP's use of windows of different sizes to select subregions in images and then determine if each subregion contains an anomaly. However, MMCI significantly reduces the computational overhead incurred by WinCLIP when simultaneously inputting dozens of images selected by windows into the CLIP's Image Encoder. Specifically, we design convolutional kernels of different sizes and shapes to process patch features extracted by the CLIP Image Encoder in parallel. Subsequently, we aggregate these features and compute their similarity with position-enhanced textual features. Through this approach, our MMCI module can effectively handle anomalies of different sizes and shapes, greatly enhancing the model's ability to localize anomaly regions.

Let $n$ different shaped convolutional kernels be denoted as $Conv_j$, where $j$ ranges from 1 to $n$. Given patch features $P_i \in \mathbb{R}^{H_i W_i \times C}$, position-enhanced text features $[F_n, F_a] \in \mathbb{R}^{2 \times C}$, normal map $M_i^n \in \mathbb{R}^{H \times W}$ and anomaly map $M_i^a \in \mathbb{R}^{H \times W}$ can be calculated by Eq. (2):

$$M_i^n, M_i^a = Up(Norm(\sum_{j=1}^{n} softmax(Conv_j(P_i) \cdot [F_n, F_a]^T))), \quad (2)$$

where $Up(\cdot)$ denotes the upsampling operation, and $Norm(\cdot)$ represents the normalization operation, ensuring that the values in the anomaly map lie between 0 and 1. By summing and normalizing $M_i$ for each layer, we can obtain the normal and anomaly map:

$$M^n = Norm(\sum_i M_i^n), \ M^a = Norm(\sum_i M_i^a), \quad (3)$$

and the final localization result can be calculated by Eq (4):

$$M = G_\sigma(M^a + 1 - M^n)/2, \quad (4)$$

where $G_\sigma$ is a Gaussian filter, and $\sigma$ controls smoothing.

## 3.4 Adapter

We employ a common bottleneck structure Adapter to align global image features and text features, consisting of two linear layers, one ReLU [15] layer, and one SiLU [12] layer, as shown in Algorithm 1.

## 3.5 Loss Functions

To learn the content of adaptive text templates and the convolutional kernel parameters in MMCI, we chose different loss functions for training from the perspectives of global anomaly detection and local anomaly localization.

---

**Algorithm 1** Adapter Module

---

**Require:** Input vector $\mathbf{x} \in \mathbb{R}^{768}$
**Ensure:** Output vector $\mathbf{y} \in \mathbb{R}^{768}$
 1: $\mathbf{h}_1 = \text{ReLU}(\mathbf{W}_1\mathbf{x} + \mathbf{b}_1) \in \mathbb{R}^{384}$
 2: $\mathbf{y} = \text{SiLU}(\mathbf{W}_2\mathbf{h}_1 + \mathbf{b}_2)$

---

*3.5.1 Global Loss.* We employ cross-entropy loss to optimize our global anomaly score as follows:

$$L_{global} = L_{ce}(S_{global}, Label), \quad (5)$$

where $S_{global}$ represents the global anomaly score calculated in Sec 3.2.2, and *Label* denotes the label indicating whether the image is anomalous or not.

*3.5.2 Local Loss.* We employ two commonly used loss functions in semantic segmentation tasks: Focal loss [21] and Dice loss [27], to optimize our anomaly map $M$, as shown by Eq. (6):

$$L_{local} = L_{focal}(M^a, gt) + L_{dice}(M^a, gt) + L_{dice}(M^n, 1 - gt). \quad (6)$$

where $gt$ is the ground truth value of anomaly maps.

## 4 Experiments

### 4.1 Datasets

Our experiments primarily focus on two datasets: MVTec [2] and VisA [39]. MVTec [2] is one of the most widely used industrial anomaly detection datasets, containing 5354 images of both normal and abnormal samples from 15 different object categories, with resolutions ranging from $700 \times 700$ to $1024 \times 1024$ pixels. VisA [39] is an emerging industrial anomaly detection dataset comprising 10821 images of normal and abnormal samples covering 12 image categories, with resolutions around $1500 \times 1000$ pixels. Similar to APRIL-GAN [5] and AnomalyCLIP [38], we conduct supervised training on the test set of one dataset and directly performed zero-shot testing on the other dataset.

### 4.2 Evaluation Metrics

Following existing AD methods [6, 35], we employ the Area Under the receiver operating Characteristic (AUC) as our evaluation metric, with image-level and pixel-level AUC used to assess anomaly detection and anomaly localization performance, respectively.

### 4.3 Implementation Details

We utilize the publicly available CLIP-L/14@336px model as our backbone, with frozen parameters for CLIP's Text Encoder and Image Encoder. Training is conducted on either the MVTec or VisA dataset, with zero-shot testing performed on the other dataset. For intermediate-level patch-based image features, we employ features from the 6-th, 12-th, 18-th, and 24-th layers of the CLIP Image Encoder. Starting from the 6-th layer, both QKV Attention and V-V Attention results are simultaneously utilized, where the outputs of QKV Attention are aligned with text features through a simple linear layer, and the outputs of V-V Attention are inputted into the MMCI module for multi-scale, multi-shape deep interaction with text features. During training, input images are resized to a

| Method | Backbone | Anomaly Description | VisA | | MVTec-AD | |
|---|---|---|---|---|---|---|
| | | | Image-AUC | Pixel-AUC | Image-AUC | Pixel-AUC |
| CLIP [29] | ViT-L/14@336px | normal / anomalous | 66.4 | 46.6 | 74.1 | 38.4 |
| CLIP-AC [29] | ViT-L/14@336px | normal / anomalous | 65.0 | 47.8 | 71.5 | 38.2 |
| WinCLIP [17] | ViT-B/16@240px | state ensemble | 78.1 | 79.6 | **91.8** | 85.1 |
| APRIL-GAN [5] | ViT-L/14@336px | state ensemble | 78.0 | 94.2 | 86.1 | 87.6 |
| AnomalyCLIP [38] | ViT-L/14@336px | normal / damaged | 82.1 | 95.5 | 91.5 | 91.1 |
| AnomalyCLIP- | ViT-L/14@336px | normal / damaged | 81.7 | 95.0 | 90.8 | 89.5 |
| **FiLo (ours)** | ViT-L/14@336px | fine-grained description | **83.9** | **95.9** | 91.2 | **92.3** |

**Table 1: Comparison results between FiLo and other ZSAD methods. The best-performing method is in bold.**

| Setup | VisA | MVTec |
|---|---|---|
| CLIP baseline | (65.0, 47.8) | (71.5, 38.2) |
| + generic [state] | (65.4, 83.9) | (79.9, 83.5) |
| **+ fine-grained [anomaly class]** | **(71.2, 85.5)** | **(80.8, 83.8)** |

**Table 2: Ablation results of anomaly descriptions. Results are displayed in the format of (Image-AUC, Pixel-AUC).**

| Setup | VisA | MVTec |
|---|---|---|
| CLIP baseline | (65.0, 47.8) | (71.5, 38.2) |
| + learnable template | (72.5, 93.1) | (82.1, **85.2**) |
| **+ fine-grained description** | **(78.1, 93.2)** | **(85.8**, 85.1) |

**Table 3: Ablation results of text template. Results are displayed in the format of (Image-AUC, Pixel-AUC).**

resolution of $518 \times 518$, and the AdamW [24] optimizer is used to optimize model parameters for 15 epochs. The learning rate for learnable text vectors is set to 1e-3, while the learning rate for the MMCI module is set to 1e-4. After that, we train the adapter for 5 epochs with a learning rate of 1e-5. Additionally, due to the varying number of fine-grained anomaly descriptions for each item category, training is conducted with a batch size of 1. Following previous methods [35, 38], a Gaussian filter with $\sigma = 4$ is applied to obtain a smoother anomaly score map during testing.

## 4.4 Main Results

To demonstrate the effectiveness of our FiLo, we compare FiLo with several existing ZSAD methods, including CLIP [29], CLIP-AC [29], WinCLIP [17], APRIL-GAN [5], and AnomalyCLIP [38]. Following [38], for CLIP, we conduct experiments using simple text prompts *A photo of a normal [class]*. and *A photo of an anomalous [class]*, and we add more text prompt templates that are recommended for ImageNet dataset for CLIP-AC. Results for Win-CLIP [17], APRIL-GAN [5], and AnomalyCLIP [38] are adopted from their respective papers. Specifically, AnomalyCLIP [38] incorporates additional learnable embeddings in the CLIP Text Encoder, while other methods, including FiLo, directly use the frozen parameters of CLIP. To ensure fair comparison, we reproduce AnomalyCLIP without learnable embeddings, referred as AnomalyCLIP-.

Table 1 presents the experimental results of FiLo and existing methods on the VisA and MVTec datasets, which demonstrates superiority of FiLo across most metrics on both datasets, validating the effectiveness of our FG-Des and HQ-Loc modules. Compared to the state-of-the-art ZSAD method AnomalyCLIP [38], after introducing the FG-Des and HQ-Loc modules, FiLo achieves a 1.1% improvement in image-level AUC and a 0.4% improvement in pixel-level AUC on the VisA dataset. Additionally, FiLo also achieves a 1.2% improvement in pixel-level AUC on the MVTec dataset.

## 4.5 Ablation Study

We conduct extensive ablation experiments on the VisA and MVTec datasets, confirming the efficacy of every component in our approach. Table 2, Table 3 and Table 4 present the experimental results of FiLo on the MVTec and VisA datasets.

In Table 2, we initially employ the same setup as CLIP-AC as our baseline, using simple two-category texts *A photo of a normal [class]* and *A photo of an anomalous [class]*. Upon realizing that the simple words "normal" and "anomalous" alone did not effectively distinguish between normal and abnormal samples, we modify the sentence structure to *A photo of a [state] [class]*, where [state] encompasses some generic descriptions for normal (e.g., perfect, flawless) and abnormal (e.g., damaged, defective) states, and observe a significant performance improvement with the introduction of more detailed [state] descriptions. Subsequently, we utilize LLMs to generate more fine-grained [anomaly class] for each class of items, resulting in further performance enhancements. This experiment underscores the effectiveness of fine-grained anomaly descriptions.

In Table 3, also starting from the CLIP baseline, we first replace all parts of the text except for [class] with learnable vectors, i.e., [v1][v2]...[vn][class]. We find that compared to handcrafted text, the text vectors learned by the model are more suitable for anomaly detection tasks, exhibiting higher detection and localization accuracy. Further, by combining the learned text vectors with detailed anomaly descriptions generated by LLMs as described earlier, we utilize the text prompt [v1][v2]...[vn][state][class] with [anomaly class], resulting in significant improvements.

In Table 4, we experiment with each component of HQ-Loc. From the table, it can be observed that both Grounding and Position Enhancement contribute to improvements in pixel-level AUC. Additionally, the MMCI module, which integrates multi-shape and multi-size capabilities, can effectively detect anomalies of various

| Grounding | Position Enhancement | MMCI | | VisA | | MVTec | |
|---|---|---|---|---|---|---|---|
| | | Multi-shape | Multi-scale | Image-AUC | Pixel-AUC | Image-AUC | Pixel-AUC |
| | | | | 78.1 | 93.2 | 85.8 | 85.1 |
| ✓ | | | | 78.1 | 93.4 | 85.8 | 85.3 |
| ✓ | ✓ | | | 78.6 | 93.6 | 85.5 | 85.7 |
| ✓ | ✓ | ✓ | | 79.2 | 95.3 | 86.2 | 89.4 |
| ✓ | ✓ | | ✓ | 80.7 | 95.6 | 88.9 | 91.4 |
| ✓ | ✓ | ✓ | ✓ | **83.9** | **95.9** | **91.2** | **92.3** |

Table 4: The results of ablation experiments for each proposed modules in HQ-Loc.

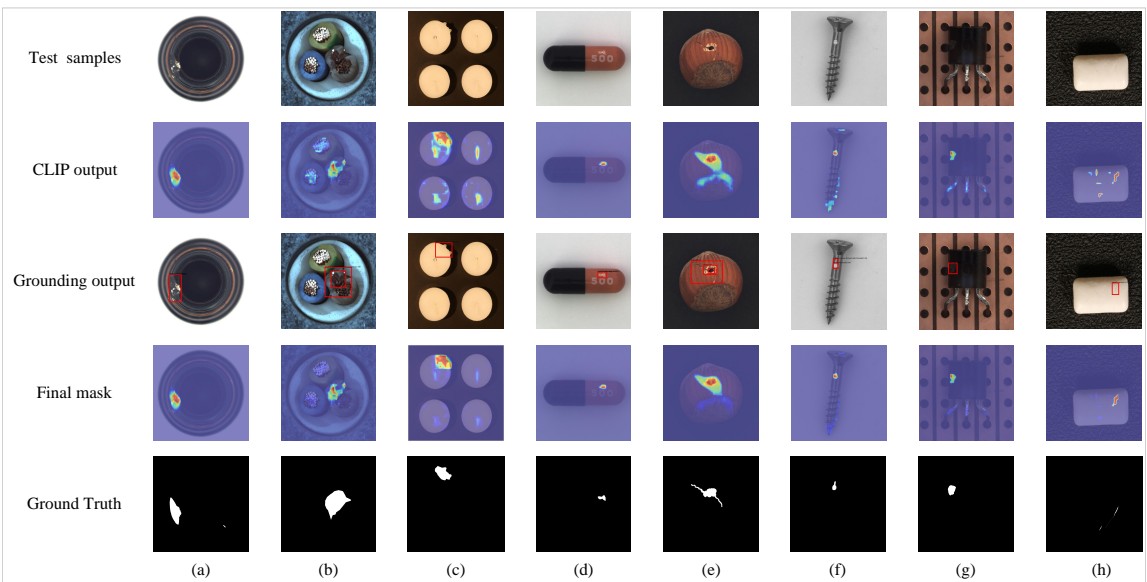

Figure 3: Visualization result of FiLo on MVTec and VisA datasets. "CLIP output" refers to the localization results without HQ-Loc, while "Final mask" represents the final localization result.

sizes and shapes, resulting in performance enhancements in both detection and localization aspects.

## 4.6 Visulization Results

Figure 3 illustrates the visualization results of our FiLo on the MVTec and VisA datasets. In the absence of any prior access to data from the target dataset, FiLo can achieve anomaly localization results that closely resemble the ground truth, showcasing FiLo's robust ZSAD capability.

As observed in the second row of Figure 3, directly computing the similarity between all patch features extracted using CLIP and textual features representing normal and abnormal semantics often yields imprecise anomaly localization results. This approach sometimes leads to false positives in non-anomalous objects or background regions of the image. However, by employing HQ-Loc's grounding for preliminary localization and position enhancement, the final output effectively mitigates this phenomenon.

Furthermore, during the preliminary localization process, Grounding associates each bounding box with matched textual descriptions, indicating the type of anomaly present in that area. For instance, in

Figure 3(e), the corresponding text for the bounding box accurately identifies two anomalies on the hazelnut: "hole" and "crack."

## 5 Conclusion

Our FiLo method represents a significant advancement in the field of Zero-Shot Anomaly Detection (ZSAD), effectively addressing prevalent challenges in both anomaly detection and localization. Our FG-Des method harnesses the capabilities of Large Language Models (LLMs) by generating specific descriptions for potential anomaly types associated with each object category. This approach notably enhances both the precision and interpretability of anomaly detection. Furthermore, our devised HQ-Loc strategy effectively mitigates the deficiencies of existing methods in terms of anomaly localization accuracy, particularly demonstrating superior performance in localizing anomalies of various sizes and shapes. Extensive experiments validate the superiority of FiLo across multiple datasets, affirming its efficacy and practicality in the realm of zero-shot anomaly detection tasks.

## Acknowledgments

This work was supported by National Key R&D Program of China under Grant No.2023ZD0120400, and National Natural Science Foundation of China (62276260, 62176254, 62076235).

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
