# OpenReview forum: "FiLo: Zero-Shot Anomaly Detection by Fine-Grained Description and High-Quality Localization"
_acmmm.org/ACMMM/2024/Conference — MM2024 Poster_

### Official Review · Reviewer_gBRw · 2024-05-03

**Rating:** 2
**Confidence:** 4

**Summary:**

The paper introduces FiLo, a novel method for Zero-Shot Anomaly Detection (ZSAD), which aims to identify anomalies in images without access to any known normal or abnormal samples within the target categories. The paper addresses the challenge of detecting anomalies in a zero-shot setting, where the model must generalize to new categories of objects not seen during training. FiLo uses Large Language Models (LLMs) to generate fine-grained anomaly descriptions for each object category, enhancing the accuracy and interpretability of anomaly detection by replacing generic "abnormal" descriptions with specific anomaly content. The paper demonstrates that FiLo significantly improves the performance of ZSAD in both detection and localization tasks. It achieves state-of-the-art results with an image-level AUC of 83.9% and a pixel-level AUC of 95.9% on the VisA dataset.

**Strengths:**

1. FiLo introduces a novel two-component approach that addresses the challenges of zero-shot anomaly detection and localization by using fine-grained descriptions and high-quality localization techniques.
2. The paper demonstrates significant improvements over existing methods, achieving state-of-the-art results on benchmark datasets, which speaks to the effectiveness of the proposed method.

**Limitations:**

1. The abstract of the paper states that FG-Des introduces fine-grained anomaly descriptions for each category using Large Language Models (LLMs). However, in the experiments, only CLIP was utilized to validate the effectiveness of the proposed method. Therefore, it is necessary for the paper to demonstrate the efficacy of the proposed approach across other LLMs, such as LLaMA, to ensure the generalizability of the findings.
2. The paper has only presented simple defects, with more complex anomalies, such as 'cable swap' and 'wire missing' in the cable category, not being showcased. There is a need to present a broader range of challenging examples to fully assess the capabilities of the proposed method.
3. The performance of the SAA method [3] is not depicted in Table 1, and the image-level AUC of the proposed method on the MVTec-AD dataset has not surpassed that of WinCLIP. This omission raises significant questions about the validity of the claims made by the authors regarding the effectiveness of their method.
4. The paper proposes the use of Grounding DINO's output to determine the position within the input text. It would be beneficial to explore whether employing bounding box coordinates or more precise defect location descriptions as inputs could enhance the method's accuracy and localization capabilities.
5. Given the strong generalization capabilities of LLMs, the paper, which leverages these models, should also demonstrate the practical utility of the proposed method in real-world defect detection scenarios or its ability to generalize across datasets.

In summary, the paper presents a novel approach to zero-shot anomaly detection using fine-grained descriptions generated by LLMs. While the results are promising, there are several areas where further validation and exploration are required to strengthen the claims and ensure the robustness of the method in various anomaly detection contexts.

**Suitability:**

2

---

### Official Review · Reviewer_vK5G · 2024-05-19

**Rating:** 3
**Confidence:** 4

**Summary:**

The paper addresses the paradox in weakly supervised video sentence grounding using mask-reconstruction, where masking aids cross-modality alignment but disrupts video-text dependency consistency. The authors propose the Mask-consistent Cross-modality Dual Reconstruction (MCDR) network to bridge relationships before and after masking, ensuring consistent reconstruction by minimizing divergences and using unmasked modality information.

**Strengths:**

1. The proposed Mask-consistent Cross-modality Dual Reconstruction (MCDR) network effectively addresses the inherent paradox in mask-reconstruction methods by maintaining consistent cross-modality dependency.
2. The effectiveness of the proposed method is demonstrated through experiments on two widely used benchmarks.

**Limitations:**

1. The performance of the proposed method is not convicing for the increment of performance is little. The compared methods are old. More up-to-date methods should be inclued, i.e., [1].
2. The proposed methods increase computational overhead. The computational overhead analyse with exisiting methods should be provided to evalutate the efficiency of the proposed method.
3. The performance of the dataset QV-Highlight should be added for more performance analyse.
4. The figure of main architecture is disordered and hard to follow.

[1] Kong, Shuhan, et al. "Dynamic Contrastive Learning with Pseudo-samples Intervention for Weakly Supervised Joint Video MR and HD." Proceedings of the 31st ACM International Conference on Multimedia. 2023.

**Suitability:**

3

---

### Official Review · Reviewer_9fS1 · 2024-05-24

**Rating:** 4
**Confidence:** 3

**Summary:**

This paper proposes a new ZSAD method called FiLo, introducing two major components: FG-Des and HQ-Loc. FG-Des uses LLMs to generate fine-grained anomaly descriptions instead of the previous general text descriptions. HQ-Loc performs anomaly localization using Grounding DINO and improves accuracy through position-enhanced text prompts and the MMCI module.

**Strengths:**

- The proposed fine-grained text descriptions and localization proposed a promising way to leverage LLM to AD task and improve the explainability and localization of current ZSAD methods.
- The text prompt is generated by LLM which reduces the manual annotation cost and increases generalization ability.
- Paper is well-written and easy to follow.
- The authors conduct comprehensive experiments and ablation studies to demonstrate the effectiveness of their method.

**Limitations:**

- LLM is used to generate fine-grained anomaly descriptions, while Grounding DINO is used for preliminary anomaly localization. These components are highly effective but their application in this paper is relatively straightforward, while MMCI is also a well-explored technique in other tasks, e.g. object detection. Therefore, the novelty of this paper is not clearly stated.
- Some of the latest related work is not discussed or compared, such as AnoVL[1].
- The experimental results presented in the supplementary material Table 3 shows that the proposed method does not consistently outperform other methods across all subsets.

- [1] AnoVL: Adapting vision-language models for unified zero-shot anomaly localization

**Suitability:**

3

---

### Official Review · Reviewer_VL11 · 2024-05-25

**Rating:** 3
**Confidence:** 3

**Summary:**

In this paper, the authors propose a zero-shot anomaly detection method called FiLo which comprises fine-grained description and position enhance high-quality localization module. Experiments on two public datasets validate the effectiveness of the proposed method.

**Strengths:**

1.	This paper is well motivated and the idea is reasonable.
2.	The proposed method achieves the new state-of-the-art on the MVTec and VisA datasets.

**Limitations:**

1. The proposed fine-grained description is proposed to accurately describe the diverse types of anomalies. However, the generation of the proposed fine-grained description relies not only on LLM model (GPT-4) but also on grounding DINO. Despite of introduction of LLM and the increase of computational overhead, the performance is not convincing compare to existing methods [1][2].
2. The proposed High-Quality Localization module lacks innovation. The methods in HQ-Loc like different shaped convolutional kernels and multi patches are common in computer vision tasks and anomaly detection.
3. Given that grounding DINO has the ability to localize abnormal regions, the proposed method appears somewhat redundant. The authors should provide a more comprehensive explanation justifying the inclusion of the proposed method alongside grounding DINO, highlighting its unique contributions and advantages.
4. More ablation studies are required to illustrate the effectiveness of different components of the fine-grained description.
5. Detailed implementation of the fine-grained description should be provided including the version of the GPT, and prompts templates of the LLM.
5. Figure 2 lacks clarity and requires improvement. It would be more comprehensible if the names of the variables used in the figure are included.
6. It is unnecessary to present Fig.3 which is merely a common adapter utilized by many methods. It is recommended to replace Figure 3 with an equation or a concise textual description.

[1] Jongheon Jeong, Yang Zou, Taewan Kim, Dongqing Zhang, Avinash Ravichan- dran, and Onkar Dabeer. 2023. Winclip: Zero-/few-shot anomaly classification and segmentation. In Proceedings of the IEEE/CVF Conference on Computer Vision and Pattern Recognition. 19606–19616.

[2] Qihang Zhou, Guansong Pang, Yu Tian, Shibo He, and Jiming Chen. 2023. Anoma- 1021 lyclip: Object-agnostic prompt learning for zero-shot anomaly detection. arXiv preprint arXiv:2310.18961 (2023).

**Suitability:**

3

---

### Meta-Review · Area_Chair_rCNS · 2024-07-03

**Recommendation:** Accept (Poster)
**Confidence:** 4

**Metareview:**

This paper proposed a novel zero-shot anomaly detection method that integrates fine-grained descriptions and high-quality localization using a combination of large language models and Grounding DINO. This work is evaluated on MVTec and VisA datasets, demonstrating state-of-the-art performance.

Pros:
- The integration of LLMs for generating fine-grained text descriptions and Grounding DINO for localization is somewhat novel.

- The proposed method achieves state-of-the-art performance on benchmark datasets.

Cons:
- Experiment results indicate that the method does not consistently outperform existing methods across all subsets.
- Some recent related works, such as AnoVL, are not discussed or compared.
- Only CLIP was utilized to validate the effectiveness of the proposed method. Demonstrating the efficacy of the proposed approach across other LLMs is necessary.

The rebuttal has addressed some concern, e.g. additional LLMs, and additional comparisons are included in the rebuttal. Thus reviewer gBRw increased the score and did not present particular reasons to reject. Overall, this paper improves the state-of-the-art results with reasonable innovations and is considered to be acceptable at ACM MM.